# Understanding the relation between Zika virus infection during pregnancy and adverse fetal, infant and child outcomes: a protocol for a systematic review and individual participant data meta-analysis of longitudinal studies of pregnant women and their infants and children

Annelies Wilder-Smith,[1] Yinghui Wei,[2] Thalia Velho Barreto de Araújo,[3] Maria VanKerkhove,[4] Celina Maria Turchi Martelli,[5] Marília Dalva Turchi,[6] Mauro Teixeira,[7] Adriana Tami,[8] João Souza,[9] Patricia Sousa,[10] Antoni Soriano-Arandes,[11] Carmen Soria-Segarra,[12] Nuria Sanchez Clemente,[13] Kerstin Daniela Rosenberger,[14] Ludovic Reveiz,[15] Arnaldo Prata-Barbosa,[16] Léo Pomar,[17] Luiza Emylce Pelá Rosado,[18] Freddy Perez,[19] Saulo D. Passos,[20] Mauricio Nogueira,[21] Trevor P. Noel,[22] Antônio Moura da Silva,[23] Maria Elisabeth Moreira,[24] Ivonne Morales,[14] Maria Consuelo Miranda Montoya,[25] Demócrito de Barros Miranda-Filho,[26] Lauren Maxwell,[27,28] Calum N. L. Macpherson,[22] Nicola Low,[29] Zhiyi Lan,[30] Angelle Desiree LaBeaud,[31] Marion Koopmans,[32] Caron Kim,[33] Esaú João,[34] Thomas Jaenisch,[14] Cristina Barroso Hofer,[35] Paul Gustafson,[36] Patrick Gérardin,[37,38] Jucelia S. Ganz,[39] Ana Carolina Fialho Dias,[7] Vanessa Elias,[40] Geraldo Duarte,[41] Thomas Paul Alfons Debray,[42] María Luisa Cafferata,[43] Pierre Buekens,[44] Nathalie Broutet,[45] Elizabeth B. Brickley,[46] Patrícia Brasil,[47] Fátima Brant,[7] Sarah Bethencourt,[48] Andrea Benedetti,[49] Vivian Lida Avelino-Silva,[50] Ricardo Arraes de Alencar Ximenes,[51] Antonio Alves da Cunha,[52] Jackeline Alger,[53] Zika Virus Individual Participant Data Consortium

For numbered affiliations see end of article.

**Correspondence to**
Dr Lauren Maxwell;
maxwelll@who.int,
lauren.maxwell.us@gmail.com

## ABSTRACT

**Introduction** Zika virus (ZIKV) infection during pregnancy is a known cause of microcephaly and other congenital and developmental anomalies. In the absence of a ZIKV vaccine or prophylactics, principal investigators (PIs) and international leaders in ZIKV research have formed the ZIKV Individual Participant Data (IPD) Consortium to identify, collect and synthesise IPD from longitudinal studies of pregnant women that measure ZIKV infection during pregnancy and fetal, infant or child outcomes.

**Methods and analysis** We will identify eligible studies through the ZIKV IPD Consortium membership and a systematic review and invite study PIs to participate in the IPD meta-analysis (IPD-MA). We will use the combined dataset to estimate the relative and absolute risk of congenital Zika syndrome (CZS), including microcephaly and late symptomatic congenital infections; identify and explore sources of heterogeneity in those estimates and develop and validate a risk prediction model to identify the pregnancies at the highest risk of CZS or adverse developmental outcomes. The variable accuracy of diagnostic assays and differences in exposure and outcome definitions means that included studies will have a higher level of systematic variability, a component of measurement error, than an IPD-MA of studies of an established pathogen. We will use expert testimony, existing internal and external diagnostic accuracy validation studies and laboratory external quality assessments to inform the distribution of measurement error in our models. We will apply both Bayesian and frequentist methods to directly account for these and other sources of uncertainty.

**Ethics and dissemination** The IPD-MA was deemed exempt from ethical review. We will convene a group of patient advocates to evaluate the ethical implications and utility of the risk stratification tool. Findings from these

analyses will be shared via national and international conferences and through publication in open access, peer-reviewed journals.

**Trial registration number** PROSPERO International prospective register of systematic reviews (CRD42017068915).

## INTRODUCTION

Zika virus (ZIKV) infection during pregnancy is an acknowledged cause of microcephaly and other forms of fetal brain defects and disability.[1 2] ZIKV is an arbovirus in the genus Flavivirus that is usually transmitted through the female *Aedes aegypti* mosquito. *Aedes aegypti* is also the main vector for dengue (DENV), urban yellow fever (YF) and chikungunya viruses. The Asian strain of ZIKV has been shown to replicate in the placenta and fetal brain;[3] ZIKV transmitted from mother to fetus during pregnancy may have a detrimental effect on fetal brain development.[4–6] Microcephaly, generally defined as a 2–3 SD reduction from the mean head circumference,[7 8] is caused by infections during pregnancy, maternal diet, drug abuse, genetic factors or environmental exposures.[9 10] Microcephaly (congenital or acquired) may be associated with developmental delays: intellectual, hearing and visual impairment and epilepsy.[11] The causal relation between ZIKV and a spectrum of fetal anomalies that includes microcephaly, now known as congenital Zika syndrome (CZS),[12] has been supported through several case-control;[13 14] cohort[15 16] and surveillance studies;[17] animal and cell studies[18] and through two systematic reviews of the evidence for causality that considered all study designs.[1 2] The relation between ZIKV infection during pregnancy and miscarriage (pregnancy loss <20 weeks gestation) and fetal loss (pregnancy loss ≥20 weeks gestation) is still under investigation.

Prior to the 2013–2016 epidemic waves, ZIKV infection was known clinically as a mild illness characterised by symptoms shared with other arboviruses, including maculopapular rash, headache, fever, non-purulent conjunctivitis and/or joint and muscle pain.[19] During the 2015–2016 ZIKV outbreak in Brazil, which extended to a number of other Latin American countries, there was a sharp increase in reports of microcephaly and other neonatal neurological conditions and in Guillain-Barré syndrome (GBS),[20–22] an autoimmune neurological disorder. Subsequent analysis of medical records collected during and after the 2013–2014 ZIKV outbreak in French Polynesia identified several ZIKV-linked pregnancies that had not been recorded earlier because they ended in elective abortion or stillbirth. The reanalysis of medical records indicated that the prevalence of both microcephaly and GBS had increased in the wake of the outbreak in French Polynesia.[23 24] The Pan American Health Organization (PAHO) issued a ZIKV Epidemiological Alert for Member States on 7 May 2015,[25] the Brazilian Ministry of Health (MOH) declared a national public health emergency due to the time and cluster of microcephaly cases identified in Northeastern Brazil on 12 November 2015[26] and the WHO declared that the clusters of microcephaly

### Strengths and limitations of this study

► This is one of the first applications of an individual participant data meta-analysis (IPD-MA) to address public health concerns in the context of an emerging pathogen. Lessons learnt from this IPD-MA may facilitate the formation of research collaborations to inform the public health response to future epidemics.

► By using a diversity of populations to develop and validate the risk prediction tool that identifies pregnancies at the highest risk of congenital Zika syndrome, the IPD-MA provides a real opportunity to help inform how clinicians and laboratory scientists communicate Zika virus (ZIKV) results to pregnant women and their families.

► There is a high degree of variability in the accuracy of diagnostic assays for ZIKV, coinfection and outcome ascertainment. Addressing this variability will be a challenge and ultimately a limitation of the accuracy of IPD-MA results.

► There is no gold standard diagnostic assay to detect ZIKV infection during pregnancy and few studies have been able to measure fetal infection. The statistical methods traditionally used to account for measurement error in IPD-MA need to be adapted to account for the myriad, correlated sources of uncertainty that arise in the synthesis of participant-level data from studies that arise in the context of an emerging pathogen.

and related neurological complications represented a Public Health Emergency of International Concern on 1 February 2016.[27]

ZIKV presents myriad challenges from an epidemiological, virological, diagnostic and outbreak control perspective. Diagnosing ZIKV infection is complicated by the absence of symptoms in most cases or the presence of non-specific symptoms; cross-reactivity with DENV;[28 29] the short window for diagnosing acute infection and the lack of point-of-care diagnostics.[30] Recent research suggests that the relation between ZIKV infection during pregnancy and fetopathology may vary by virus genotype or lineage; primary versus secondary infection[31] and DENV-immune status and genotype in the presence of coinfection.[29 32 33] The unequal spatial distribution of microcephaly cases has been discussed extensively.[34–36] These differences may be related to population-level differences in baseline risk of adverse fetal outcomes (clinically important heterogeneity), differences in study design (eg, inclusion criteria; measurement of important cofactors) or to measurement error, defined as the difference between the observed and actual level of a given variable. Laboratory confirmation of ZIKV infection and coinfection differs by diagnostic algorithms (eg, definition of positive and negative ZIKV diagnostic assay results); factors that affect the regularity of testing (eg, provision of incentives, distance from testing centre, differences across protocols); population-specific distribution of related coinfections; differing levels of training of laboratory staff and the accessibility of materials and technology (eg, ultrasound, immunoassays, reliability panels), among other factors. In addition to documented difficulties in accurately measuring infant head circumference, measurement standards for identifying microcephaly differ across populations

and standards themselves may not appropriately classify reduced or enlarged head circumference.[37 38]

Our limited understanding of the absolute risk of adverse fetal, infant and child outcomes in ZIKV-infected mothers led to calls from several governments suggesting that women avoid becoming pregnant for as long as 2 years.[39 40] ZIKV disproportionately affects low-income populations residing in areas with poor living conditions.[41] The impetus placed on women to delay pregnancy as a ZIKV control measure is complicated by the limited access to contraception and safe abortion in many of the countries and regions with the highest burden of ZIKV-related microcephaly.[42 43] Identifying the risk factors for CZS is a global health priority and central for prioritising resource allocation for vector control and effective and targeted family planning interventions and for improving risk counselling for ZIKV-infected pregnant women or women planning a pregnancy in endemic areas.

## Rationale for the individual participant data meta-analysis of longitudinal studies of pregnant women

Individual participant data meta-analysis (IPD-MA) is the quantitative synthesis of participant-level data from included studies, while appropriately accounting for the clustering of information at the study level. The proposed IPD-MA will combine deidentified, participant-level cohort data from different populations of pregnant women to identify and quantify the relative importance of different predictors of CZS. Individual participant data (IPD) have a number of analytic benefits over aggregate data meta-analysis (AD-MA), a form of knowledge synthesis that combines study-level measures of effect.[44 45] IPD facilitate the assessment of effect measure modification, the development and validation of risk prediction models, and the application of a unified analytic approach. In addition to using the same statistical model across studies, with IPD we can apply the same or similar exclusion criteria, diagnostic algorithms, methods for addressing missing data and confounding and conduct the same types of sensitivity analyses needed to explore unexplained within- and between-study heterogeneity.

### Increased precision of estimates

Timely, accurate and reliable predictions are predicated on well-designed studies that minimise the risk of bias, adequate sample size and the inclusion of a diversity of populations. Adequate sample size is crucial for precise estimation of the risk of CZS within important subgroups (eg, women infected during the first trimester; pregnant women with previous or concurrent DENV, CHIKV and STORCH (syphilis, toxoplasmosis, rubella, cytomegalovirus and herpes) pathogen exposure). Vector control measures, including pesticides, public education campaigns, the use of drones to detect standing water and the introduction of sterilised male vectors to reduce *Aedes aegypti* populations, have been implemented in the wake of the 2015/2016 ZIKV epidemics.[46–48] Fortunately, these measures, in combination with other factors that are currently being investigated, seem to have reduced the numbers of ZIKV infections during the 2017/2018 epidemic cycle. While many studies have followed infants to the end of their first year, certain developmental milestones can only be assessed after age 2[49] or when a child reaches school age. Leveraging limited data from studies with extended follow-up of ZIKV-infected and non-infected women will be essential for estimating the risk of more subtle, long-term effects of ZIKV infection during pregnancy. By combining data from individual studies, the proposed IPD-MA will improve the precision of risk estimates.

### Identify and quantify the relative importance of effect measure modifiers

The benefits of using IPD rather than AD to assess effect measure modification and interaction are myriad.[50] In a one-stage analysis with IPD, subject level data are meta-analysed using the exact binomial distribution; in a two-stage analysis of IPD or AD, study-level outcome measures are combined assuming asymptomatic normality.[51] In a one-stage analysis of IPD, study-level and individual-level sources of heterogeneity can be assessed concurrently and IPD are better able to identify heterogeneity in the context of rare events or small studies.[50 52] Individual studies are often powered to detect the overall effect of the exposure rather than subgroup effects. Due to variations in the characteristics of the affected populations and in the potential confounders and effect modifiers measured by different studies, it is unlikely that individual studies will be powered to definitively quantify the importance of different sources of heterogeneity in the relation between ZIKV infection during pregnancy and adverse fetal, infant or child outcomes.

### Clinical risk prediction to inform decision-making and resource allocation

While there are a number of vaccine trials underway,[53] the development of a ZIKV vaccine is complicated by the necessity of testing the vaccine in pregnant women; assessing whether the vaccine is associated with development of GBS; the difficulties inherent in developing an arbovirus vaccine;[46 54–56] findings from in vivo studies that indicate cross-reactivity between ZIKV and DENV or West Nile virus is related to antibody-dependent enhancement of ZIKV infection;[55 57 58] and by the potential use of prevention of infection as a vaccine efficacy endpoint.[59] In this context, identifying the pregnancies at the highest risk of adverse neonatal and later developmental outcomes is critical for effective resource allocation and prevention strategies. We will use participant-level data to develop and externally validate clinical risk prediction models to facilitate the identification of pregnancies that are most likely to result in ZIKV-related adverse fetal or infant outcomes and longer-term developmental delays.

### Standardisation and cross-national partnerships to inform the public health response to emerging pathogens

#### Formation of the ZIKV IPD Consortium

The ZIKV IPD Consortium is a global collaboration designed to streamline the international response to ZIKV. To facilitate cross-country analyses and a coordinated response to ZIKV, representatives from WHO, PAHO, the US Centers for Disease Control and Prevention (CDC), the National Institutes of Health (NIH), the National Institute of Allergy and Infectious Diseases (NIAID), Institut national de la santé et de la recherche médicale (INSERM), Institut Pasteur, and the networks of Fundação Oswaldo Cruz (Fiocruz), Grupo de Pesquisa da Epidemia da Microcefalia (MERG)/ZikaPlan, ZIKAlliance, ZIKAction, the Consortium for the Standardization of Influenza Seroepidemiology (CONSISE) and International Severe Acute Respiratory and Emerging Infection Consortium (ISARIC) have developed a standardised protocol for cohorts of pregnant women and their infants exposed to ZIKV to facilitate the proposed IPD-MA; identified existing or planned cohorts and prospectively introduced cohort principal investigators (PI)s and MOH officials to the methodological and public health benefits related to IPD-MA in the context of Zika. Many of the longitudinal studies and surveillance systems identified to date through the review of country-level registries, existing literature reviews and ZIKV IPD Consortium membership have agreed to contribute deidentified, participant level data to the analysis. A complete list of the studies and surveillance systems who have agreed to contribute data to the ZIKV IPD Consortium led IPD-MA is included in online supplementary table 1.

#### Standardised protocols for cohorts of pregnant women and their infants

A multiplicity of mechanisms for exposure and outcome ascertainment as well as differences in the measurement of important cofactors are known challenges for the meta-analysis of data from individual research studies. To minimise the potential for heterogeneity caused by differences in study inclusion criteria and the measurement of ZIKV, infant outcomes and important cofactors, WHO/PAHO, Institut Pasteur, Fiocruz, CONSISE and ISARIC convened an international meeting of ZIKV researchers and MOH officials in June 2016 to develop standardised protocols and data collection instruments for cohort studies of pregnant women and newborns and other ZIKV-related studies.[60] Standardisation of protocols and data collection instruments was intended to minimise differences in case ascertainment and data collection methods to facilitate data synthesis and the identification of sources of heterogeneity in the relation between congenital Zika infection and adverse fetal, infant and child outcomes. The protocols were made available on WHO website in October 2016 (http://www.who.int/reproductivehealth/zika/en). The standardised protocols do not include detailed guidance on laboratory methods, but testing algorithms were developed by an expert panel and made available on the WHO website in March 2016 (http://www.who.int/csr/resources/publications/zika/laboratory-testing/en/). The IPD-MA will need to account for the between-study and within-study differences in diagnostic assays and testing algorithms.

### OBJECTIVES OF THE IPD-MA

1. Estimate the absolute and relative risks of fetal infection; miscarriage (<20 weeks gestation), fetal loss (≥20 weeks gestation), microcephaly and other manifestations of CZS and later developmental delays for women who do and do not experience ZIKV infection during pregnancy.
2. Identify factors that modify women's risk of adverse ZIKV-related fetal, infant and child outcomes and infants' risk of infection (eg, gestational age at time of infection, clinical or subclinical illness, concurrent or prior arbovirus exposure, other congenital infections and other posited effect measure modifiers).
3. Use information on the relative importance of different effect measure modifiers identified in Objective 2 to decompose the total effect of ZIKV infection during pregnancy on adverse fetal, infant and child outcomes into (1) the direct effect of ZIKV; (2) the indirect effect of ZIKV as mediated by the effect measure modifier of interest (eg, DENV, CHIKV or STORCH pathogens) and (3) the effect of the interaction between ZIKV and the mediator of interest.
4. Develop and validate a risk prediction tool to identify pregnant women at a high risk of an adverse ZIKV-related outcome and to inform couples planning a pregnancy, healthcare providers and/or resource mobilisation (eg, vector control strategies; antenatal care; open access to contraception).

### METHODS AND ANALYSIS

This protocol has been drafted in accordance with the PRISMA-P Statement (online supplementary table 2).[61] The proposed systematic review and meta-analysis will follow the PRISMA-IPD guidelines for the systematic review of non-randomised studies.[62]

#### Step 1. Study identification

##### Eligibility criteria

Eligible studies will use a longitudinal design where ZIKV infection is measured in pregnant women prior to outcome ascertainment. Eligible studies may include cohort studies, case-cohort studies, randomised control trials or active surveillance systems, regardless of publication status. Studies may enrol symptomatic and/or asymptomatic women prior to or following a confirmed pregnancy. Included studies and active surveillance systems will test women for ZIKV infection during pregnancy, follow women until the end of pregnancy and assess for CZS or related fetal, infant or child outcomes (see table 1). We will exclude studies with fewer than 10

**Table 1** Participant-level variables of interest

| | |
|---|---|
| Exposure | Maternal ZIKV infection (diagnosis: confirmed, probable, unlikely; primary, secondary, naïve; viral load) |
| | Fetal or placental ZIKV infection (diagnosis: confirmed, probable, unlikely; primary, secondary, naïve; viral load)* |
| Primary outcomes | Miscarriage (<20 weeks gestation) |
| | Fetal loss (≥20 weeks gestation) |
| | Microcephaly (diagnosis: severe microcephaly, microcephaly, normocephaly, macrocephaly, Z-score) |
| | CZS (diagnosis: confirmed, probable, unlikely) |
| Secondary fetal outcomes† | Induced abortion with microcephaly (diagnosis: confirmed, probable, unlikely) |
| | Early fetal death (20–27 weeks gestation) |
| | Late fetal death (≥28 weeks gestation) |
| | Late fetal death (≥28 weeks gestation) with microcephaly |
| | Placental insufficiency (diagnosis: confirmed, probable, unlikely)‡ |
| | Intrauterine growth restriction |
| Secondary infant outcomes† | Postnatal microcephaly (diagnosis: severe microcephaly, microcephaly, normocephaly, macrocephaly; Z-score) |
| | Gestational age at birth |
| | Birth weight (diagnosis: normal birth weight; low birth weight; very low birth weight; extremely low birth weight; Z-score) |
| | Craniofacial disproportion |
| | Neuroimaging abnormalities (intracranial calcification, lissencephaly, hydranencephaly, porencephaly, ventriculomegaly, posterior fossa abnormalities, cerebellar hypoplasia, corpus callosal and vermian dysgenesis; focal cortical dysplasia) |
| | Postnatal intraventricular haemorrhage |
| | Motor abnormalities (hypotonia, hypertonia, hyperreflexia, spasticity, clonus, extrapyramidal symptoms)§ |
| | Seizures, epilepsy§ |
| | Ocular abnormalities (blindness, other)§ |
| | Congenital deafness or hearing loss§ |
| | Congenital contractures (arthrogryposis, unilateral or bilateral clubfoot) |
| | Other non-neurological congenital abnormalities |
| Secondary outcomes detected after the infant period¶ | Cortical auditory processing |
| | Neurodevelopment (expressive and receptive language, fine and gross motor skills, attention and executive function, memory and learning, socioemotional development, overall neurodevelopmental score) |
| | Vision (Cardiff test) |
| Posited confounders | Demographic factors (age, education, marital status, racial/ethnic group; BMI) |
| | Socioeconomic factors |
| | Maternal smoking, illicit drug and alcohol use |
| | Maternal prescription drug use, vaccination |
| | Maternal experience of violence during pregnancy; infant or child exposure to intimate partner violence[127] |
| | Workplace or environmental exposures to teratogenic substances (eg, maternal exposure to lead, mercury) |

Continued

**Table 1** Continued

| Potential effect measure modifiers | Genetic anomalies, metabolic disorders, perinatal brain injury |
| --- | --- |
| | Gestational age, term at birth |
| | Timing of infection during pregnancy |
| | Clinical/subclinical illness |
| | Viral genotype and load |
| | Concurrent or prior flavivirus or alphavirus infection |
| | Maternal history of YF or JE vaccination |
| | Maternal immunosuppressive conditions, disorders, comorbidities (eg, chronic hypertension, diabetes) or pregnancy-related conditions (eg, pre-eclampsia, gestational diabetes) |
| | Intrauterine exposure to STORCH pathogens |
| | Maternal malnutrition |
| | Presence and severity of maternal and infant clinical symptoms |

*Fetal ZIKV infection will be considered as both an exposure and an outcome; definition of fetal infection will be based on clinical and radiological criteria defined by an expert panel.
†Both with and without microcephaly.
‡As estimated by antenatal consequences of placental insufficiency, including fetal growth restriction, oligohydramnios, non-reassuring fetal heart rate tracing or small for gestational age at birth as markers of placental insufficiency.
§May also be detected after the infant period.
¶As measured by the Bayley Scale;[128] Ages and Stages;[129] INTERGROWTH-21st Neurodevelopmental Assessment.[49]
BMI, body mass index; CZS, congenital Zika syndrome; JE, Japanese encephalitis; STORCH, syphilis, toxoplasmosis, rubella, cytomegalovirus and herpes; YF, yellow fever virus; ZIKV, Zika virus

participants and limit included surveillance systems to those that capture country or territory-level active surveillance data (ie, individual hospital active surveillance data will not be included). Before sharing participant-level data, research studies will be asked to provide documentation of ethics review.

## Information sources
### ZIKV IPD Consortium

We anticipate that most eligible studies will have been identified through the efforts of the ZIKV IPD Consortium. The Consortium is an international initiative that is meant to include the PIs from all planned, ongoing or completed ZIKV longitudinal studies at the time of this review. We have searched clinical trials and ZIKV-related databases[63] (online supplementary table 3) to identify existing or planned longitudinal studies. We have circulated the list of ongoing or planned ZIKV-related longitudinal studies of pregnant women to MOH Officials in countries with autochthone ZIKV transmission and to PIs of ZIKV cohorts and asked them to update the list as necessary.

### Systematic review

We will perform a systematic search of biomedical databases for published longitudinal studies and protocols. The search strategy is based on Medical Subject Headings (MeSH) and text-based search terms for ZIKV, pregnant women, infants and children. The search strategy was developed in collaboration with an information scientist and adapted for the following electronic databases: Embase (Medline), Embase (Ovid) and SCOPUS (see online supplementary text 1 for the search strategy for Embase (Medline and Ovid). We also will search the additional databases listed in online supplementary table 3 and review the reference lists of published systematic reviews and the list of studies produced by a living systematic review of ZIKV studies conducted by the University of Bern[64] to identify additional studies. After removing duplicates from the list of identified studies, two reviewers will independently screen the title and abstracts of included studies to identify longitudinal studies or active surveillance systems that measure ZIKV infection during pregnancy and subsequent fetal, infant or child outcomes. Disagreements about study inclusion will be resolved by consensus.

### Collection of study-level data

We will contact the PIs of eligible studies identified through either the ZIKV IPD Consortium or the electronic searches to invite them to take part in the IPD-MA and ask them to provide a copy of their study protocol. We will develop and pilot an electronic data extraction form to record study-level characteristics for all eligible studies, regardless of whether study PIs agree to participate in the IPD-MA. Two reviewers will independently review protocols and study-related publications to extract data on study design; study population; enrolment, follow-up and laboratory procedures; assay and specimen type; criteria used to define ZIKV infection and timing of infection and exposure, cofactor and outcome ascertainment for all eligible studies. We will ask study PIs for clarification if there are outstanding questions or disagreements regarding study-level data.

### Step 2. Collection, review and synthesis of deidentified, participant-level data

We will contact the PIs and authors of studies that meet our inclusion criteria to request deidentified, participant-level data on select variables and the associated surveys and data dictionaries or codebooks. If study data have been imputed, we will request both the original and imputed data so that we can apply consistent imputation methods across studies and review the imputed dataset for validation purposes. To reduce the burden on individual studies and ensure clear documentation of all steps in the creation of the synthesised dataset, we will use the study codebooks or data dictionaries to develop study-specific code in the statistical language used by the study data manager that selects only the study variables required for the proposed analyses and removes information that could be used to identify individual participants. The study's data manager will apply the code to the original dataset. The deidentified, participant-level data will be transferred from the study site to Emory University, which will serve as the WHO data synthesis partner centre, using secure file transfer protocol and will be protected on a secure server with standard encryption and by the Emory University firewall. Data synthesis-related decisions will be reviewed by a ZIKV IPD Consortium membership and will be recorded using Jupyter Notebook.[65] Researchers who are unable or unwilling to provide their participant data after at least four attempts at contact by the project team over a period of 6 months will be excluded from the IPD-MA and we will report the reason for their exclusion. When IPD are not available for a given study, we will extract study-level effect estimates from any publications to compare study-level estimates from all eligible studies, whether or not they provide data for the IPD-MA.

### Variables of interest

Despite efforts to develop protocols that can be applied across studies, there will be significant cross-study heterogeneity in how congenital Zika infection, cofactors and outcomes are measured and reported. Exposure, outcome variables and posited confounders and effect measure modifiers are listed in table 1. Where possible, ZIKV and other infections (eg, DENV, CHIKV, STORCH pathogens) will be modelled as time-varying, rather than time-fixed covariates. Given that the case definitions for microcephaly have changed over time (and may change during the course of included studies), we will allow for the coding of variables with different definitions (ie, WHO fetal growth chart,[66] Fenton scale,[67] INTERGROWTH 21st Project[49]). We will ask studies for data on the continuous measures used to make diagnoses (eg, viral load;

head circumference) rather than just the diagnoses themselves (eg, maternal ZIKV infection, microcephaly). Using continuous variables will allow us to test the sensitivity of results to the application of different cutoffs and the reference standards used to generate Z-scores. Definitions for miscarriage, fetal loss and other pregnancy outcomes vary across countries. We will explore the sensitivity of project findings to different outcome definitions.

### Assessing the integrity of deidentified, participant-level data
We will review the distribution of variables to identify potential outliers and to assess the proportion missing within each study. We will discuss the distribution of key variables with the study data manager to identify and address any inconsistencies. If there has been a publication related to a given longitudinal study, we will attempt to replicate table 1 presented in the publication and will resolve any inconsistencies with the data manager.

### Synthesis of participant-level data
Given that these longitudinal studies and active surveillance systems are part of the global research response to an emerging pathogen, there is a high degree of variability in the data that have been collected across studies and the algorithms that have been applied to define ZIKV exposure, symptoms, components of CZS and so on. Where possible, we will ask studies for the individual factors (ie, fever, rash) that were used to define certain parameters (ie, clinical infection) to ensure cross-study consistency in composite markers. Similarly, we will combine the data inputs for exposure, cofactor and outcome classification algorithms to reduce cross-study differences in the classification of important factors.

### Critical review of study quality
We will use the Cochrane Methodological Quality Assessment of Observational Studies[68] and the Q-Coh tool[69] to help describe the risk of bias within non-randomised studies and will apply the Cochrane Risk of Bias 2.0 tool to assess the risk of bias in randomised controlled trials.[70] Rather than using a score-based bias assessment, a panel that includes experts on the evaluation of laboratory assays and external quality assessment (EQA); obstetrics and perinatal epidemiology will provide a detailed description of the role of selection, confounding and measurement-related biases within studies.

## Step 3. Statistical analyses
### Objectives 1 and 2
*Estimate the absolute and relative risks of adverse ZIKV-related fetal, infant and child outcomes; identify and quantify relative importance of sources of heterogeneity*

Estimating the absolute risk of CZS by the gestational age of the fetus at the time of infection is as important as it is difficult. Early in the outbreak, cohort studies limited enrolment to symptomatic pregnant women. While an estimated 50%–70% of infections are subclinical, when symptoms are detected they generally appear 3–14 days after infection.[71] For asymptomatic infections, the gestational age of infection is interval censored because it is defined by the last negative and first positive tests for ZIKV. Rather than using the midpoint between the last negative and first positive ZIKV test, which is known to be biased, we will impute the trimester or week that asymptomatic infections occurred using methods that are routinely applied in studies with interval censored covariates in the field of perinatal research.[72 73] In table 2, we present sample definitions for the absolute risk of fetal and infant outcomes. These definitions will be reviewed prior to analysis and publication and we will assess the sensitivity of our results to the definition applied. Later developmental outcomes (eg, neurodevelopment, cortical auditory processing), listed in table 1 as secondary outcomes, will follow a fetuses-at-risk approach.[74] We will apply censoring to account for competing risks where necessary.

We will apply mixed binomial models for binary outcomes, and multinomial models for categorical outcomes, with a logit link to provide estimates for each measure of absolute risk by week or trimester of congenital infection. Because of the differences in baseline risks across populations, pooling measures of absolute risk

| Table 2 | Definitions applied to estimation of absolute risk of primary fetal and infant outcomes | |
|---|---|---|
| **Outcome** | **Numerator** | **Denominator** |
| Miscarriage | Number of miscarriages (pregnancy loss prior to 20 weeks gestation) | Total number of pregnancies |
| Early fetal death | Number of pregnancies lost between 20 and 27 weeks gestation | Total number of pregnancies carried to 20 weeks gestation |
| Late fetal death | Number or pregnancies lost at or following 28 weeks gestation | Total number of pregnancies carried to 28 weeks gestation |
| Microcephaly | Number of microcephaly cases | Total number of pregnancies carried to ≥24 weeks gestation, when microcephaly can be assessed by ultrasound in ZIKV-infected mothers,[38] we will consider all pregnancies regardless of whether the pregnancy results in a live birth |

ZIKV, Zika virus.

across studies may not be clinically relevant and can even be misleading.[75] We will combine study-level estimates of absolute risk through: (1) a one-stage meta-analysis (mixed binomial or multinomial model with a log link) that includes study-level sources of heterogeneity and a separate intercept for each study to account for additional cross-study differences in baseline risk and (2) a forest plot of study-level estimates of absolute risk that does not include a summary meta-analytic estimate.

Absolute measures of effect are considered more important for informing clinical practice than relative measures.[76] We will conduct both (1) a one-stage meta-analysis where we estimate the relative risk of the aforementioned outcomes of interest by congenital Zika infection across studies and (2) a two-stage meta-analysis where we estimate the relative risk in each study and combine study-level measures using random effects meta-analysis to allow the underlying true effect to vary across studies.[77] In the one-stage models, we will include study-specific intercepts to quantify and account for between-study variation in baseline risk. We will use random slopes to allow the relation between certain cofactors and the risk of CZS to vary across populations.

Combining absolute measures of effect, like the risk difference, across studies may mask important differences in the baseline risk.[78] We will present estimates of the risk difference in a forest plot of study-level estimates without presenting a summary meta-analytic estimate. In both the one-stage and two-stage analyses, we will use log binomial regression models to estimate the relative risk of each binary outcome and will use log Poisson regression to estimate the relative risk if log binomial models fail to converge.[79 80] In the two-stage models, we will assess the potential for non-linear relationships between continuous exposures (viral load) and covariates (eg, gestational age, maternal age) by using the Akaike information criteria to compare restricted cubic splines with three knots to exponential, quadratic and linear terms. In the one-stage models, we will use generalised additive mixed models (GAMMs) to assess potential non-linearities as the GAMM random smoothing parameter addresses the bias/variance trade-off by penalising the added complexity from non-linear terms while accounting for between-study variation in non-linear effects.[81]

### Joint estimation of multiple nested or otherwise related outcomes (multivariate meta-analysis)

Not all studies will have measured all primary or secondary outcomes of interest. For example, most studies will have measured ventriculomegaly, but may not include values for intracranial calcification or ocular abnormalities.[9] This analysis is intended to increase the precision of estimates of the spectrum of CZS abnormalities. Studies that do not include the measurement of a given outcome will necessarily be excluded from univariate estimates of that outcome, but will be included in multivariate models that estimate the joint probability of related outcomes. In the multivariate models, we will assume that the outcomes that

are excluded from certain studies are missing at random and will incorporate studies by setting the missing observations and within-study correlations between outcomes to zero and will set the within-study variance to a very high number such that the artificial value that acts as a substitute for the missing outcome will have a negligible effect on the meta-analytic estimate from the multivariate model.[82] Alternatively, under a Bayesian framework, we will model a joint distribution for studies providing multiple outcomes and a univariate distribution for studies providing a single outcome without needing to address the missing within-study correlations and variance for studies with only one outcome.[83] The secondary outcomes that will be included in the multivariate analysis are listed in table 1.

We will compare generalised linear mixed models (GLMMs) where we use one model to analyse nested or otherwise related outcomes to the standard univariate approach where we apply a separate model to analyse each outcome. Multivariate meta-analysis allows for the estimation of joint probabilities across multiple outcomes and accounts for cross-study and within-study correlation between related outcomes.[82 84] Modelling several outcomes simultaneously improves the precision over univariate models by sharing information about heterogeneity and the average effect of the treatment which may facilitate inference about the relation between different CZS-related outcomes[82 85 86] (ie, vermian dysgenesis and ocular abnormalities).

### Multivariate model to combine estimates from fully and partially adjusted studies

A number of longitudinal studies will not include the minimal sufficient set of confounders. Estimates from partially adjusted studies (that are missing values for important confounders) will be combined with fully adjusted estimates in a one-stage multivariate meta-analysis. The one-stage multivariate model allows us to borrow information from partially adjusted studies with different sets of confounders while ensuring that we control for important confounders.[82 85]

### Special considerations for the meta-analysis of cohort studies with rare events

Two-stage meta-analytic methods are based on large sample approximations and may be unsuitable in the context of CZS, which can be considered a rare event.[87 88] Two-stage meta-analysis may be biased when small studies are included, the effect of an exposure is very large or the outcome is rare, all of which may affect this analysis.[89] We will highlight any instances when the two-stage meta-analytic estimates may be biased by the aforementioned issues and will limit our inference to one-stage analyses in those cases. If we have a number of longitudinal studies with zero events, we will focus our inference on a one-stage approach to avoid reliance on large sample approximations.

### Assessment of study-level and participant-level heterogeneity

Separating within-study and between-study heterogeneity is central to assessing participant-level heterogeneity and to understanding the relative importance of different potential effect measure modifiers.[50] We are only able to separate within-study and between-study heterogeneity across studies that include both levels of the effect measure modifier of interest. The presence of clinical illness may be related to disease course through viral load or be a marker for the strength of the immune system's response to infection. We will conduct a one-stage analysis of longitudinal studies that include both symptomatic and asymptomatic women to assess whether the risk of CZS or of the most severe effects of congenital infection (miscarriage, fetal loss) differs for clinical and subclinical infections. Between-study heterogeneity is reflective of study-level differences, while within-study heterogeneity may be indicative of clinically important differences. We will mean centre covariates included in the interaction terms at the study level to separate between-study and within-study heterogeneity in our one-stage meta-analytic estimates of how prior or co-infection with alpha or flaviviruses or STORCH pathogens modifies the effect of ZIKV infection.[90]

Heterogeneity in effect estimates will arise from clinically important differences between congenital infections or women (effect measure modification) and from study-level differences in exposure and outcome ascertainment (measurement error). With IPD, we are able to jointly assess study-level and participant-level heterogeneity.[52] We will incorporate participant-level interaction terms in a one-stage analysis that includes random intercepts to account for unmeasured study-level factors. We will consider random slopes for certain covariates to allow for between-study variation in covariate effects across studies. Given the difficulty in assessing the total df in mixed models, we will apply bootstrapping to assess the approximate confidence intervals of the pooled interaction terms. We will present the analysis of effect measure modifiers in accordance with the revised STROBE guidelines.[91]

Based on our review of research protocols for planned or ongoing cohort studies, we expect to include data from longitudinal studies with different enrolment criteria, exposure and outcome ascertainment, diagnostic assays for prior-infections or coinfections and measurement of important cofactors. We will include measures of study-level sources of heterogeneity (eg, diagnostic assay, outcome definitions) as covariates in the one-stage regression to assess the variance explained by these factors. We will perform a sensitivity analysis where we limit our inference to studies with similar inclusion criteria and exposure, cofactor and outcome ascertainment to reduce spurious cross-study heterogeneity. While two-stage analyses of interaction effects that fail to separate between-study and within-study heterogeneity are subject to ecological bias[90] and our inference about the importance of interaction terms will primarily be derived from one-stage analyses, we will use a two-stage analysis to compare the magnitude of the interaction effects across studies. The interaction between certain cofactors and ZIKV exposure may not be consistent across studies. In the first stage of the two-stage analysis, we will use the likelihood ratio test (p<0.05) to assess the importance of including interaction terms within each study. Individual cohort studies may not have the sample size needed to detect clinically important interactions between ZIKV and important cofactors. We will also assess whether a certain interaction is consistent across studies, while not necessarily statistically significant within individual studies.

Meta-regression and subgroup analyses have limited power to detect interactions and can only be used to make inference about the relation between the exposure and study-level, average values of participant characteristics.[89 92] Studies that are not willing or able to provide participant-level data may differ importantly from longitudinal studies whose data is included in the IPD-MA. We will apply subgroup analysis to a two-stage analysis of effect estimates from studies included in the IPD-MA and published estimates from studies that did not participate in the IPD-MA to assess whether study-level variation in recruitment and enrolment criteria, exposure and outcome ascertainment and measurement of coinfections and other cofactors are important sources of heterogeneity in the pooled estimates. Some sources of heterogeneity (eg, vector density and feeding patterns; DENV serotype) may not be measured and should be considered in sensitivity analyses.

### Objective 3

*Use information on the relative importance of different effect measure modifiers identified in Objective 2 to decompose the total effect of ZIKV infection during pregnancy on adverse fetal, infant and child outcomes*

Some studies suggest that antibody-dependent enhancement related to concurrent or prior DENV infection or Japanese encephalitis vaccination may modify the effect of ZIKV infection on fetal development. Both the timing of exposure to DENV and DENV serotype may contribute to regional differences in the strength of the relation between ZIKV infection and CZS.[28 32] If we find evidence in the literature that the effect measure modifier identified in Objective 2 (eg, DENV) may affect the outcome (eg, CZS), we will apply inverse probability of treatment weighted-marginal structural models to decompose the total effect of ZIKV on the outcome of interest into the direct effects of ZIKV infection, the effect of ZIKV infection mediated by the posited effect measure modifier and the effect of the interaction between ZIKV and the effect measure modifier.[93 94]

### Objective 4

*Develop and validate a risk prediction tool to inform decision making by pregnant women, couples planning a pregnancy and healthcare providers and/or resource mobilisation*

We will fit one-stage logistic regression models with random intercepts to account for differences in the

baseline risk within each study. We will apply group Lasso regression[95] to identify the prognostic variables that predict progression to miscarriage, fetal loss and microcephaly. Lasso regression is implemented using L1-penalised estimation. The application of group Lasso ensures that the algorithm selects all levels of categorical variables by treating corresponding dummy variables as a group instead of allowing the model to only select certain levels of categorical variables.[96 97] The L-1 penalty term allows for concurrent consideration of predictors and shrinkage, which facilitates variable selection in the context of high dimensional data.[98] We will standardise included variables so that all variables use the same scale. We will adopt cross-validation on the study level to select the optimal tuning parameter ($\lambda$) and will adopt restricted maximum likelihood (REML) to estimate the variance-covariance matrix of the study-level random effects.

Not all studies will have the resources to implement the most accurate and reliable ZIKV-related diagnostic tools. As part of the data synthesis, we will identify the exposure and cofactor diagnostic methods that are most commonly applied. As a sensitivity analysis, we will use these diagnostic methods to develop a risk prediction model so that the model can be applied in regular clinical practice.

### Development and external validation of the prediction model

We will apply internal-external cross-validation where we rotate the cohort that is used for external validation to improve the model's predictive ability.[99] For example, given k cohort studies, we will use k-1 cohort studies to develop the prediction model and will validate model performance by applying the prediction model to a cohort that was not used to develop the prediction model. Internal-external cross-validation allows for the use of all available data for model development and validation which improves model performance and generalisability.[100]

### Evaluation of model performance

We will generate receiver operating characteristic (ROC) curves[101 102] in the cohort that was not used to develop the prediction model to estimate the model's true-positive (sensitivity) versus false-positive (1-specificity) rate for each binary outcome. These curves will then be summarised using the area under the ROC curve (AUC). In some instances, the pregnant woman or couple planning a pregnancy may prefer a more sensitive rather than a more specific model. We will present a range of cut-off values that maximise sensitivity, specificity or both sensitivity and specificity to facilitate decision making by pregnant women or couples planning a pregnancy. We will assess the extent to which these thresholds yield consistent sensitivity and specificity across different regions and populations. We will use calibration plots to compare the observed and predicted probability of the outcome of interest within risk quintiles and summarise these plots by calculating the total ratio of observed versus expected events (O:E ratio) and the calibration slope. Internal-external cross-validation of k studies will result in kAUCs,

O:E ratios and calibration slopes. We will apply random effects meta-analysis to combine estimates of the discrimination and calibration of the kpredictive models. We will assess model calibration and discrimination and choose the model with the best properties.[99 103] We will use bootstrap validation to evaluate model optimism and will follow the TRIPOD statement guidelines for reporting the final prediction models.[104]

### Step 4. Quantitative bias analysis

Given the complexity and level of measurement error, we will conduct a quantitative bias analysis under a Bayesian framework where we use a combination of expert opinion, laboratory EQA and external and internal assessment of the relative accuracy of diagnostic assays and other methods for cofactor and outcome ascertainment to inform the prior distributions of bias parameters. Where possible, we will apply frequentist methods for quantitative bias analysis[105] as a sensitivity analysis and will use the GRADE criteria[106] to compare the quality of the evidence from Bayesian and frequentist models, with a focus on how imprecision, inconsistency, indirectness, magnitude of effect differ in the Bayesian and frequentist approaches to addressing the myriad sources of bias expected to affect these analyses.

### Selection bias

Studies or surveillance systems that only recruit or test symptomatic pregnant women or studies that only enrolled pregnant women who tested positive for ZIKV infection are affected by selection bias because selection into the study is associated with the exposure.[63] This situation is similar to the inclusion of a single treatment arm in a randomised controlled trial. Although data from studies that only enrol pregnant women who test positive for ZIKV cannot directly inform estimates of the causal effect of ZIKV, these data can inform the development of prediction models because they contain information on the prognosis of ZIKV positive women. Longitudinal studies that restrict enrolment to ZIKV positive pregnant women may also increase the precision of relative treatment effects by providing more events within ZIKV-exposed pregnant women. Longitudinal studies have reported that women who perceive their infants as unaffected by CZS are less likely to participate in follow-up. We will consider matching on the propensity score or the use of inverse probability of censoring weights[107] and prognostic score analysis[108] to account for measured determinants of differential loss to follow-up in the aetiological and prognostic models, respectively. Selection bias can be induced when we inappropriately adjust for a time-varying confounder affected by prior exposure (a confounder that also acts to mediate the relation between ZIKV infection and adverse fetal, infant or child outcomes). We will use G-computation methods to appropriately adjust for time-dependent confounders affected by prior exposure.[109]

## Confounding bias

We will adjust for confounders that are unlikely to mediate the causal relation between infection during pregnancy and adverse infant outcomes (table 1). We will estimate each participant's likelihood of being infected during pregnancy, conditional on the study group and important confounders, to identify possible violations of the positivity assumption. In sensitivity analyses, we will apply propensity score matching within studies to ensure that important confounders are adequately balanced across exposure groups. Despite the prospective, collaborative development of a standardised research protocol for ZIKV cohort studies of pregnant women, confounders and effect measure modifiers may be defined differently across studies or not measured in certain studies. We will develop a detailed codebook that reflects the heterogeneity in confounder definitions and report on this heterogeneity in our analyses.

## Measurement (ie, detection, misclassification) bias

Despite efforts to harmonise case definitions across studies with the prospective development of a standardised protocol for cohorts of pregnant women and their infants,[60] the case definitions, diagnostic tools and algorithms used to ascertain ZIKV infection, cofactors and CZS-associated outcomes vary across studies.[110] The literature on the accuracy of ZIKV-related and DENV-related assays is evolving rapidly.[30 111] Prior to initiating our analyses, we will synthesise the current evidence on the sensitivity and specificity of different assays for ZIKV diagnosis, for the assessment of concurrent or prior DENV infections and for estimating the time of infection, among other relevant factors. The WHO standardised protocol for ZIKV-related cohorts of pregnant women includes WHO recommendations on the screening and assessment of neonates and infants with intrauterine ZIKV exposure;[112] we will compare study-level outcome definitions with the standardised WHO definitions. The role of heterogeneity related to case definitions and diagnostic tools will be explored through both frequentist and Bayesian methods. In the frequentist approach, we will: (1) include categorical or continuous markers of sensitivity and specificity of diagnostic tools as study-level covariates in the one-stage analyses and (2) apply diagnostic tool specific-subgroup analysis to both the one-stage and two-stage meta-analysis of effect measures from different studies. In the Bayesian approach, we will use a combination of expert opinion and data from external and internal validation studies to inform the probability distributions of bias parameters.[113]

## Missing data

Missing data at the study level, as when confounders are not measured in certain studies, is a well-known challenge of IPD-MA[114 115] and a likely source of residual confounding. In keeping with current recommendations for addressing missingness in IPD-MA, we will apply new methods for multilevel multiple imputation to account for missing values.[116] As a sensitivity analysis, we will impute missing participant-level data in each study separately and use multivariate meta-analysis to combine data across studies that have and have not measured important host-level and environmental-level cofactors.

## Publication bias

IPD-MA may have a lower risk of publication bias than AD-MA because they include data from unpublished studies.[114] We have tried to ensure that the ZIKV IPD Consortium includes representatives from all of the academic and government institutions responsible for planned or ongoing ZIKV-related longitudinal studies of pregnant women and their infants. We expect that Consortium members will identify most ZIKV longitudinal studies and active surveillance systems of pregnant women and their infants, regardless of publication status, and we will conduct a systematic review to identify additional longitudinal studies and active surveillance systems. The degree of publication bias will be assessed visually by reviewing the asymmetry of study-level estimates from published and unpublished studies using funnel plots that compare log RR to the corresponding studies' sample size.[117]

We will convene a group of patient advocates to evaluate the ethical implications and utility of the risk stratification tool.

## DISCUSSION

The application of IPD-MA to an emerging pathogen presents an important opportunity to harness global collaboration to inform the development of recommendations for pregnant women, couples planning a pregnancy and public health practitioners. While IPD-MA offers real benefits compared with AD-MA or to the inference possible with individual cohort studies, the ability of IPD-MA to inform public health practice is directly related to the quality of the exposure, cofactor and outcome ascertainment in the original cohort studies. Statistical methods for IPD-MA were developed in the context of clinical research and randomised control trials. These methods needs to be adapted to account for the myriad sources of uncertainty and bias that affect observational research, especially for field epidemiology studies conducted as part of the research response to unknown or emerging pathogens.

Historically, arboviruses and other neglected tropical diseases have been understudied because the burden of disease falls on under resourced populations in the Global South.[118] In the context of ZIKV, the unequal distribution of risk is coupled with inequities in access to preventative measures like modern contraception and to critical clinical and therapeutic care for infants affected by microcephaly and ZIKV-related neurological disorders. Each case of microcephaly is associated with a loss of 29.95 disability adjusted life years (DALYs) and treatment costs ranging from US$91K to US$1 million.[119] To put these figures into perspective, the yearly per capita

income in Pernambuco, the Brazilian state with one of the highest burdens of CZS, is US$3471.[120]

There is no vaccine for ZIKV and the only treatment is supportive.[58] There have been numerous calls for data sharing[121][122] and cooperation between governments and academic institutions,[54][123] and public and private charities have pledged significant financial support to improve our understanding of ZIKV epidemiology and to develop a vaccine or small molecule prophylaxis to decrease the risk of infection. In the wake of the Ebola epidemic, the global response to ZIKV has been characterised by unprecedented levels of international cooperation. In the absence of a ZIKV vaccine or prophylaxis, international leaders in ZIKV research have formed the ZIKV IPD Consortium to identify, collect and synthesise IPD from longitudinal studies of pregnant women that measure ZIKV infection during pregnancy and fetal, infant and child outcomes. These data will be used to quantify the absolute risk of ZIKV-related pregnancy complications with the goal of aiding women and their families in making difficult reproductive decisions and with helping public health systems prevent and quantify the burden of congenital Zika infection.

### Challenges of developing and conducting an individual participant data-meta-analysis in the context of an emerging pathogen

Ideally, researchers prespecify confounders, effect measure modifiers and plans for subgroup or sensitivity analyses in their research protocol. In the context of Zika, our understanding of the virus is changing so rapidly that analysis plans may change significantly despite our best efforts to review the latest evidence on transmission, immunological response, diagnostic assays, vector biology and basic ZIKV epidemiology. Our ability to appropriately account for measurement error will play a critical role in the accuracy of estimates for the risk of CZS and other adverse fetal, infant and child outcomes. This is one of the first instances where an IPD-MA has been used to address public health concerns in the context of an emerging pathogen. We expect that best practices and lessons learnt from this IPD-MA can be used to facilitate the formation of research collaborations to streamline the public health response to future epidemics.

### Patient and public involvement

In keeping with guidelines for public involvement in research,[124] knowledge users (ie, women of reproductive age and their families, clinicians) will be consulted at each stage of this research. The research question and protocol were designed with feedback from clinicians who treat pregnant women in ZIKV-endemic areas and infants and children affected by CZS. Focus groups that include women of reproductive age in ZIKV-endemic areas will be used to evaluate the ethical implications and utility of the risk stratification tool in three countries.

### ETHICS AND DISSEMINATION

This IPD-MA protocol has been deemed exempt from ethical review by the WHO Ethics Review Committee and the Emory University Institutional Review Board. Individual longitudinal studies will provide documentation of ethics review prior to sharing their deidentified, participant-level data. The WHO has developed guidance for data sharing in public health emergencies or in the context of emerging pathogens.[125] Sharing deidentified data for IPD-MA is generally considered exempt from ethical review if the objectives of the IPD-MA are in keeping with the objectives of the original studies.[126] Individual research studies and consortia will secure additional ethics review and/or legal guidance on the sharing of deidentified, subject-level data as needed. The results of this analysis will be published under the ZIKV IPD Consortium name and will include a list of the names of key investigators from each study that contributed data for that analysis and researchers who contributed to the analysis or writing at the end of the publication. Findings from the proposed analysis will be shared via national and international conferences; existing platforms for dissemination of ZIKV-related research (eg, The Global Health Network) and through publication in open access, peer-reviewed journals.

**Author affiliations**

[1]Lee Kong Chian School of Medicine, Nanyang Technological University, Singapore, Singapore
[2]Centre for Mathematical Sciences, University of Plymouth, Plymouth, UK
[3]Department of Social Medicine, Universidade Federal de Pernambuco, Recife, Brazil
[4]Health Emergencies Programme, Organisation mondiale de la Sante, Geneve, Switzerland
[5]Department of Collective Health, Institute Aggeu Magalhães (CPqAM), Oswaldo Cruz Foundation, Recife, Brazil
[6]Institute of Tropical Pathology and Public Health, Federal University of Goias, Goiânia, Brazil
[7]Department of Biochemistry and Immunology, Federal University of Minas Gerais, Belo Horizonte, Minas Gerais, Brazil
[8]Department of Medical Microbiology, University Medical Center Groningen, Groningen, The Netherlands
[9]Department of Social Medicine, University of São Paulo, São Paulo, Brazil
[10]Reference Center for Neurodevelopment, Assistance, and Rehabilitation of Children, State Department of Health of Maranhão, Sao Luís, Brazil
[11]Department of Pediatrics, University Hospital Vall d'Hebron, Barcelona, Spain
[12]SOSECALI C. Ltda, Guayaquil, Ecuador
[13]Department of Epidemiology, University of São Paulo, São Paulo, Brazil
[14]Department of Infectious Diseases, Section Clinical Tropical Medicine, UniversitatsKlinikum Heidelberg, Heidelberg, Germany
[15]Evidence and Intelligence for Action in Health, Pan American Health Organization, Washington, District of Columbia, USA
[16]Department of Pediatrics, D'Or Institute for Research & Education, Rio de Janeiro, Brazil
[17]Department of Obstetrics and Gynecology, Centre Hospitalier de l'Ouest Guyanais, Saint-Laurent du Maroni, French Guiana
[18]Hospital Materno Infantil de Goiânia, Goiânia State Health Secretary, Goiás, Brazil
[19]Communicable Diseases and Environmental Determinants of Health Department, Pan American Health Organization, Washington, District of Columbia, USA
[20]Department of Pediatrics, FMJ, São Paulo, Brazil
[21]Faculdade de Medicina de Sao Jose do Rio Preto, Department of Dermatologic Diseases, São José do Rio Preto, Brazil
[22]Windward Islands Research and Education Foundation, St. George's University, True Blue Point, Grenada

23 Department of Public Health, Universidade Federal do Maranhão – São Luís, São Luís, Brazil

24 Department of Neonatology, Oswaldo Cruz Foundation (Fiocruz), Rio de Janeiro, Brazil

25 Facultad de Salud, Universidad Industrial de Santander, Bucaramanga, Colombia

26 Faculty of Medical Sciences, University of Pernambuco, Recife, Brazil

27 Reproductive Health and Research, World Health Organization, Geneva, Switzerland

28 Hubert Department of Global Health, Emory University, Atlanta, Georgia, USA

29 Institute of Social and Preventive Medicine, University of Bern, Bern, Switzerland

30 McGill University Health Centre, McGill University, Montréal, Canada

31 Pediatric Infectious Diseases, Stanford Hospital, Palo Alto, California, USA

32 Department of Virology, Erasmus Medical Center, Rotterdam, The Netherlands

33 Department of Reproductive Health and Research, World Health Organization, Geneva, Switzerland

34 Department of Infectious Diseases, Hospital Federal dos Servidores do Estado, Rio de Janeiro, Brazil

35 Instituto de Puericultura e Pediatria Martagão Gesteira, Universidade Federal do Rio de Janeiro, Rio de Janeiro, Brazil

36 Statistics, University of British Columbia, British Columbia, Vancouver, Canada

37 INSERM CIC1410 Clinical Epidemiology, CHU La Réunion, Saint Pierre, Réunion

38 UM 134 PIMIT (CNRS 9192, INSERM U1187, IRD 249, Université de la Réunion), Universite de la Reunion, Sainte Clotilde, Réunion

39 Children's Hospital Juvencio Matos, São Luís, Brazil

40 Sustainable Development and Environmental Health, Pan American Health Organization, Washington, District of Columbia, USA

41 Department of Gynecology and Obstetrics, University of São Paulo, São Paulo, Brazil

42 Julius Center for Health Sciences and Primary Care, University Medical Center Utrecht, Utrecht, The Netherlands

43 Mother and Children Health Research Department, Instituto de Efectividad Clinica y Sanitaria, Buenos Aires, Argentina

44 School of Public Health and Tropical Medicine, Tulane University, New Orleans, USA

45 Department of Reproductive Health and Research, World Health Organization, Geneva, Switzerland

46 Department of Infectious Disease Epidemiology, London School of Hygiene and Tropical Medicine, London, UK

47 Instituto de pesquisa Clínica Evandro Chagas, Fundacao Oswaldo Cruz, Rio de Janeiro, Brazil

48 Facultad de Ciencias de la Salud, Universidad de Carabobo, Valencia, Carabobo, Bolivarian Republic of Venezuela

49 Departments of Medicine and of Epidemiology, Biostatistics & Occupational Health, McGill University, Montreal, Quebec, Canada

50 Department of Infectious and Parasitic Diseases, Faculdade de Medicina da Universidade de Sao Paulo, São Paulo, Brazil

51 Department of Tropical Medicine, Federal University of Pernambuco, Recife, Brazil

52 Department of Pediatrics, Federal University of Rio de Janeiro, Rio de Janeiro, Brazil

53 Facultad de Ciencias Médicas, Universidad Nacional Autónoma de Honduras, Tegucigalpa, Honduras

**Acknowledgements** The authors would like to acknowledge the valuable contributions of Liège Maria Abreu de Carvalho, Rosangela Batista, Ana Paula Bertozzi, Gabriel Carles, Denise Cotrim, Luana Damasceno, Lady Dimitrakis, María Manoela Duarte Rodrigues, Cassia F Estofolete, Maria Isabel Fragoso da Silveira Gouvêa, Vicky Fumadó-Pérez, Rosa Estela Gazeta, Neely Kaydos-Daniels, Suzanne Gilboa, Amy Krystosik, Véronique Lambert, Milagros García López-Hortelano, Marisa Marcia Mussi-Pinhata, Christina Nelson, Karin Nielsen, Denise M Oliani, Renata Rabello, Marizelia Ribeiro, Barry Rockx, Laura C. Rodrigues, Silvia Salgado, Katia Silveira, Elena Sulleiro, Van Tong, Diana Valencia, Wayner Vieira de Souza, Luis Angel Villar Centeno and Andrea Zin to the review of the ZIKV IPD Consortium IPD-MA Protocol. In addition, the authors would like to acknowledge the BMJ Open reviewers for their thoughtful comments on the protocol.

**Collaborators** Liège Maria Abreu de Carvalho; Rosangela Batista; Ana Paula Bertozzi; Gabriel Carles; Denise Cotrim; Luana Damasceno; Lady Dimitrakis; María Manoela Duarte Rodrigues; Cassia F. Estofolete; Maria Isabel Fragoso da Silveira Gouvêa; Vicky Fumadó-Pérez; Rosa Estela Gazeta; Neely Kaydos-Daniels; Suzanne Gilboa; Amy Krystosik; Véronique Lambert; Milagros García López-Hortelano; Marisa Marcia Mussi-Pinhata; Christina Nelson; Karin Nielsen; Denise M. Oliani; Renata Rabello; Marizelia Ribeiro; Barry Rockx; Laura C. Rodrigues; Silvia Salgado; Katia Silveira; Elena Sulleiro; Van Tong; Diana Valencia; Wayner Vieira de Souza; Luis Angel Villar Centeno; Andrea Zin.

**Contributors** NB, CBH, TJ, NL, LM, JS and LR contributed to the initial conception of the study. AB, TD, PG, NL, LM and YW made substantial contributions to the statistical methodology proposed for the IPD-MA. LM wrote the first draft of the protocol. AW-S, YW, TVBA, MV, CMTM, MDT, MT, AT, PS, JPS, AS-A, CSS, AMS, NSC, KDR, LR, APB, LP, LEPR, FP, SP, MN, TN, MEM, IM, MCMM, DBMF, LM, CM, NL, ZL, ADL, MK, CK, EJ, TJ, CH, PG, PG, JG, ACFD, VE, GD, TPAD, MLC, PB, NB, EB, PB, FB, SB, AB, VAS, RAAX, AAC and JA provided substantial revisions to the protocol. All authors approved the final version of the protocol.

**Funding** The development of the IPD-MA protocol was supported by a Wellcome Trust grant to the WHO Department of Reproductive Health and Research Human Reproduction Programme, grant number 206532/Z/17/Z.

**Competing interests** None declared.

**Patient consent for publication** Not required.

**Provenance and peer review** Not commissioned; externally peer reviewed.

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
