## [Reviewer comments · BMJ Open]

ARTICLE DETAILS

TITLE (PROVISIONAL)	Understanding the relation between Zika virus infection during pregnancy and adverse fetal, infant, and child outcomes: a protocol for a systematic review and individual participant data meta-analysis of longitudinal studies of pregnant women and their infants and children
AUTHORS	Zika Virus Individual Participant Data Consortium, -; Wilder-Smith, Annelies; Wei, Yinghui; Velho Barreto de Araújo, Thalia; VanKerkhove, Maria; Turchi Martelli, Celina Maria; Turchi, Marília Dalva; Teixeira, Mauro; Tami, Adriana; Souza, João; Sousa, Patricia; Soriano-Arandes, Antoni; Soria-Segarra, Carmen; Sanchez Clemente, Nuria; Rosenberger, Kerstin Daniela; Reveiz, L; Prata-Barbosa, Arnaldo; Pomar, Léo; Pelá Rosado, Luiza Emylce; Perez, Freddy; Passos, Saulo; Nogueira, Mauricio; Noel, Trevor P.; Moura da Silva, Antônio; Moreira, Maria Elisabeth; Morales, Ivonne; Miranda Montoya, Maria Consuelo; Miranda-Filho, Demócrito de Barros; Maxwell, Lauren; Macpherson, Calum; Low, Nicola; Lan, Zhiyi; LaBeaud, Angelle Desiree; Koopmans, M; Kim, Caron; João, Esaú; Jaenisch, Thomas; Hofer, C. B.; Gustafson, Paul; Gérardin, Patrick; Ganz, Jucelia S; Fialho Dias, Ana Carolina; Elias, Vanessa; Duarte, Geraldo; Debray, Thomas; Cafferata, Maria Luisa; buekens, pierre; Broutet, Nathalie; Brickley, Elizabeth B.; Brasil, Patrícia; Brant, Fátima; Bethencourt, Sarah; Benedetti, Andrea; Avelino-Silva, Vivian; Arraes de Alencar Ximenes, Ricardo; Alves da Cunha, Antonio; Alger, Jackeline

VERSION 1 - REVIEW

REVIEWER	Dr. Jordan Cates Division of Congenital and Developmental Disorders, National Center on Birth Defects and Developmental Disabilities, CDC, USA; Oak Ridge Institute for Science and Education, Oak Ridge, Tennessee I declare that that I have read and understood the BMJ Group policy on declarations of interest. I hereby declare the following interests, according to the policy. Organizational: I work as a research fellow on the Zika en Embarazadas y Niños (ZEN) cohort study that may be included in this IPD-MA.
REVIEW RETURNED	17-Sep-2018

GENERAL COMMENTS	Overall The authors thoroughly describe a detailed protocol for assessing the impacts of congenital Zika virus (ZIKV) infection on fetal, infant, and child outcomes using an individual participant data (IPD) meta-analysis (MA) framework. This work is strengthened by the established collaborations through the ZIKV IPD Consortium. Overall this is a well written manuscript with a strong analytic
--

approach. I applaud the authors for their detailed description of the proposed methodology as well as the considerations of the limitations inherent in IPD-MA and the unique limitations likely to be faced with pooling data from studies designed during an emerging outbreak (e.g. evolving diagnostics). A few considerations for the authors are described below.

Considerations:

Strengths and limitations of this study, page 7: "The statistical methods traditionally used to synthesize IPD across clinical studies and randomized controlled trials of need to be adapted to account for these myriad sources of uncertainty." Throughout the description of the methods in pages 17-24 there was no mention of adapting current statistical methods. Either the methods section needs to be clarified to articulate what is being 'adapted', or this sentence should be rewritten in such a way that it doesn't suggest that a statistical method is being adapted but rather a statistical method that incorporates these sources of uncertainty will be adopted.

Methods & Analysis

Eligibility criteria, page 12: Authors state that included studies will test women for ZIKV infection during pregnancy. While there was a lot of description about the potential variability in diagnostics and definitions, there was no description of potential differences in protocol as it relates to how often women are tested for ZIKV. Since this has huge implications for the primary exposure I believe it is worth mentioning.

ZIKV IPD Consortium, page 13: Will analyses described in this paper have to be postponed until the individual results of each study are published? Has this been discussed with collaborators? If some agreement has been discussed I would recommend mentioning this in the manuscript.

Variables of interest, page 14: The third sentence of this paragraph suggests that authors intend to use the codings adopted by individual studies for microcephaly. I would recommend obtaining raw head circumference data from all studies and analyzing them using a consistent definition; it is unclear based on this sentence whether or not that is already the plan or if the plan is to use the coding for microcephaly that was used by the individual studies.

Variables of interest, Table 1, page 14-16: After the secondary infant outcomes section, the variables of interest listed no longer have any analytic definition (binary vs categorical etc.). I realize this is likely because there are many different ways to define all of these outcomes, confounders, and effect measure modifiers, but it is inconsistent with the earlier sections of the table. I would recommend either adding these definitions in or adding an explanation as to why they are not all defined.

Statistical analyses

Objectives 1&2, Table 2, page 17: The denominators in Table 2 are very thorough and appear to align with a fetuses at risk approach. However, no outcomes are listed in this table for child outcomes such as later developmental delays. I realize these are considered secondary outcomes, but it might be worth mentioning whether the authors plan to consider a fetuses at risk approach for these child outcomes as well, potentially using censoring for competing risks, or

	whether those analyses will be restricted to live births (https://obgyn.onlinelibrary.wiley.com/doi/full/10.1111/aogs.13194). Objectives 1&2, page 18: In the paragraph starting with ‘Combining absolute measures of effect...’, please add a reference for the first sentence. Objectives 1&2, page 18: In the second to last sentence, the authors state that studies with missing outcomes will have their outcomes set to zero with a very high within-study variance so that the substitution for the missing outcome ‘will have a negligible effect on the meta-analytic estimate’. If this will have a negligible effect on the meta-analytic estimate by design, then why even include it in the model? Assessment of study- and participant-level heterogeneity, page 20: The second paragraph on page 20 (‘Based on our review...’) does not have a single reference. There are some statements, such as, but not limited to, ‘two-stage analyses of interaction effects are subject to ecological bias...’ that warrant a reference. Also in this paragraph, for the last sentence it is unclear how the authors plan to assess whether a certain interaction is consistent across studies but not statistically significant within studies; this should be clarified. Objective 3, page 21: I have some concerns about the third objective looking at decomposing the total effect of ZIKV infection using causal mediation analysis. The primary objective is to decompose the total effect of ZIKV infection on infant outcomes via mediation through concurrent or prior DENV infection. The assumption behind this analysis is that DENV infection is along one causal pathway from ZIKV infection to infant outcomes (ZIKV → DENV → Infant outcome). However, the authors do not mention any evidence that DENV infection itself could cause an adverse fetal/infant/child outcome. I believe that prior evidence warrants assessing DENV infection as an effect measure modifier, but given the current explanation I am not convinced that it warrants mediation assessment. Additionally given that there is some evidence of Zika-Dengue interaction due to antibody-dependent enhancement, there may be conceptual issues related to a possible mediator-exposure interaction that are not incorporated into standard approaches of mediation (https://www.ncbi.nlm.nih.gov/pmc/articles/PMC5476432/). Selection bias, page 23: The authors mention potential selection bias related to inappropriately adjusting for time-varying confounding. However, there was no mention earlier of assessing ZIKV infection as a time-varying exposure. This should be added to the section on exposure definitions. General comments Could the authors investigate risk factors for ZIKV infection using the pooled data? There was no mention of this objective in the manuscript. Also, is the consortium open to other proposals for research questions not specified in this manuscript? If so, that should be articulated.
--	--

REVIEWER	Jaime E. Castellanos
	Laboratorio de Virologia Universidad El Bosque, Bogota Colombia
REVIEW RETURNED	23-Sep-2018

GENERAL COMMENTS	The ethics approval of each participant should be guaranteed by the PI of each study through a certificate they ask each of the parents for the permission. It is important to state specifically how the researchers could integrate and/or compare the prediction models or outcomes of each study with those obtained by this IPD study. Do deserve the serology system used in primary studies more restriction?. Current studies use different home made or commercial techniques, and they could introduce differences in prevalence numbers, mainly in those areas where dengue circulate or where YFV vaccination is mandatory.
--

REVIEWER	Adriani Nikolakopoulou Institute of Social and Preventive Medicine (ISPM), University of Bern, Switzerland
REVIEW RETURNED	05-Nov-2018

GENERAL COMMENTS	This is a protocol for an individual participant data meta-analysis to estimate the risk of Zika virus infection during pregnancy, examine potential sources of heterogeneity and develop a risk prediction model to identify pregnancies at highest risk of congenital Zika syndrome. It is a collaborative work aiming to include the PIs from all relevant longitudinal studies. The planned analysis is described clearly and in detail and the importance of the review is evident. I have only a few comments, mainly pertaining to the methodology. Major comments 1) Authors discuss the important differences in the distribution of microcephaly cases and aim to identify and explore heterogeneity. What will authors' strategy be in the case of severe heterogeneity? Will they still estimate the overall relative and absolute risk of CZS or will they only present results per subgroup? 2) Authors state that "IPD can be analyzed in either a one- or a two-stage meta-analysis while AD can only be meta-analyzed using a two-stage approach" (Page 13 of 45, lines 24-25). I think that this statement might be misleading as one-stage meta-analysis can also be used when only AD are available. See for example model (2) in the paper by Simmonds and Higgins (Simmonds and Higgins, 2016) where the exact likelihood is used instead of making a normality assumption for the effect estimates. Simmonds, M.C., and Higgins, J.P. (2016). A general framework for the use of logistic regression models in meta-analysis. Stat. Methods Med. Res. 25, 2858–2877. 3) Authors say that they will use GRADE to compare quality of evidence in Bayesian and frequentist models (Page 25 of 45, lines 41-45). To my understanding, the only model that will be applied both in a Bayesian and a frequentist framework is the multivariate meta-analysis (described in Page 21 of 45, lines 41-end). Could you clarify to which models quality of evidence will be assessed? Also, why the focus is on comparing quality in Bayesian and frequentist models instead of assessing quality of evidence for the key analyses? What would a different assessment depending on the framework imply?
---

VERSION 1 – AUTHOR RESPONSE

RESPONSE TO REVIEWER 1

Reviewer #1: The authors thoroughly describe a detailed protocol for assessing the impacts of congenital Zika virus (ZIKV) infection on fetal, infant, and child outcomes using an individual participant data (IPD) meta-analysis (MA) framework. This work is strengthened by the established collaborations through the ZIKV IPD Consortium. Overall this is a well written manuscript with a strong analytic approach. I applaud the authors for their detailed description of the proposed methodology as well as the considerations of the limitations inherent in IPD-MA and the unique limitations likely to be faced with pooling data from studies designed during an emerging outbreak (e.g. evolving diagnostics). A few considerations for the authors are described in the attached file.

Strengths and limitations of this study, page 7: “The statistical methods traditionally used to synthesize IPD across clinical studies and randomized controlled trials of need to be adapted to account for these myriad sources of uncertainty.” Throughout the description of the methods in pages 17-24 there was no mention of adapting current statistical methods. Either the methods section needs to be clarified to articulate what is being ‘adapted’, or this sentence should be rewritten in such a way that it doesn’t suggest that a statistical methods is being adapted but rather a statistical method that incorporates these sources of uncertainty will be adopted.

The reviewer makes an important point here. Our principal intention was to suggest that methods for data synthesis and cross-study analysis in the context of IPD-MA, developed for clinical studies and randomized controlled trials, need to be adapted to address the myriad sources of uncertainty and correlated error terms that arise the conduct of IPD-MA methods in the context of field epidemiology studies of an emerging pathogen. We speak about the quantitative bias analysis on page 23. The language in the Strengths and Limitations section has been modified to state:

- There is no gold standard diagnostic assay to detect ZIKV infection during pregnancy and few studies have been able to measure fetal infection. The statistical methods traditionally used to account for measurement error in IPD-MA need to be adapted to account for the myriad, correlated sources of uncertainty that arise in the synthesis of participant-level data from studies that arise in the context of an emerging pathogen.

Methods & Analysis

Eligibility criteria, page 12: Authors state that included studies will test women for ZIKV infection during pregnancy. While there was a lot of description about the potential variability in diagnostics and definitions, there was no description of potential differences in protocol as it relates to how often women are tested for ZIKV. Since this has huge implications for the primary exposure I believe it is worth mentioning.

The reviewer is correct. There are tremendous differences between studies in terms of how and how often pregnant women are tested for ZIKV. Please note that, on page 10 of the protocol, we speak to these differences:

Laboratory confirmation of ZIKV infection and co-infection differs by diagnostic algorithms (e.g. definition of positive and negative ZIKV diagnostic assay results); factors that affect the regularity of testing (e.g. provision of incentives, distance from testing center, differences across protocols); population-specific distribution of related co-infections; differing levels of training of laboratory staff; and the accessibility of materials and technology (e.g., ultrasound, immunoassays, reliability panels), among other factors. In addition to documented difficulties in accurately measuring infant head circumference, measurement standards for identifying microcephaly differ across populations and standards themselves may not appropriately classify reduced or enlarged head circumference.^{37 38}

ZIKV IPD Consortium, page 13: Will analyses described in this paper have to be postponed until the individual results of each study are published? Has this been discussed with collaborators? If some agreement has been discussed I would recommend mentioning this in the manuscript.

Principal investigators had quite different opinions about how to approach the timing of the IPD-MA-related papers. No consensus was reached when the issue was discussed at the initial meeting of cohort investigators in February, 2017. Part of the utility of an IPD-MA is to include studies that have not yet published or may not plan to publish their results. This will be particularly important for the IPD-MA as many country- and territory-level active surveillance systems may not plan to publish their findings.

We added the following text to the paragraph on page 13 for clarification:

Eligible studies may include cohort studies, case-cohort studies, randomized control trials, or active surveillance systems, regardless of publication status.

Discussions on when and how to share data will continue to evolve as new studies and country- or territory-level active surveillance systems are identified and agree to contribute data to the IPD-MA. The ZIKV IPD Consortium will need to explore this issue further when we are closer to the analysis phase.

Variables of interest, page 14: The third sentence of this paragraph suggests that authors intend to use the codings adopted by individual studies for microcephaly. I would recommend obtaining raw head circumference data from all studies and analyzing them using a consistent definition; it is unclear based on this sentence whether or not that is already the plan or if the plan is to use the coding for microcephaly that was used by the individual studies.

We agree wholeheartedly with this assessment. Our plan is to use infant head circumference where available from individual studies and to utilize studies' assessment of microcephaly when not

available. We have added the following text to page 15 further clarify our intention to use continuous measures (e.g. head circumference) to estimate important exposure, outcome, or covariate measures where possible.

We will ask studies for data on the continuous measures used to make diagnoses (e.g. viral load; head circumference) rather than just the diagnoses themselves (e.g. maternal ZIKV infection, microcephaly). Using continuous variables will allow us to test the sensitivity of results to the application of different cutoffs and the reference standards used to generate Zscores.

Variables of interest, Table 1, page 14-16: After the secondary infant outcomes section, the variables of interest listed no longer have any analytic definition (binary vs categorical etc.). I realize this is likely because there are many different ways to define all of these outcomes, confounders, and effect measure modifiers, but it is inconsistent with the earlier sections of the table. I would recommend either adding these definitions in or adding an explanation as to why they are not all defined.

We thank the reviewer for their close attention to detail and agree with their assessment that we were inconsistent in how we described exposure and outcome measures as compared to confounders or posited effect measure modifiers. We agree with the reviewer that the multiple ways that confounders and potential mediators can and will be described across studies make them impractical to predefine for inclusion in Table 1. We have amended Table 1 to remove the description of how the exposure and primary outcomes will be classified (i.e. binary, continuous, categorical), while leaving some of the detail related to how exposure and outcome-related diagnoses are commonly categorized. We hope that this change is sufficient to improve consistency.

Statistical analyses

Objectives 1&2, Table 2, page 17: The denominators in Table 2 are very thorough and appear to align with a fetuses at risk approach. However, no outcomes are listed in this table for child outcomes such as later developmental delays. I realize these are considered secondary outcomes, but it might be worth mentioning whether the authors plan to consider a fetuses at risk approach for these child outcomes as well, potentially using censoring for competing risks, or whether those analyses will be restricted to live births.

(<https://obgyn.onlinelibrary.wiley.com/doi/full/10.1111/aogs.13194>).

The reviewer brings up an excellent point here and we very much appreciate them pointing us to the related literature. We have added the following text to page 17:

Later developmental outcomes (e.g. neurodevelopment, cortical auditory processing), listed in Table 1 as secondary outcomes, will follow a fetuses-at-risk approach;⁷⁷ and we will apply censoring to account for competing risks.

Objectives 1&2, page 18: In the paragraph starting with 'Combining absolute measures of effect...', please add a reference for the first sentence.

We apologize for our oversight here and thank the reviewer for their close attention to detail. We have added the following reference to the text on Page 18:

Egger M, Smith GD, Phillips AN. Meta-analysis: Principles and procedures. *BMJ* 1997;315(7121):1533-37. doi: 10.1136/bmj.315.7121.1533

Objectives 1&2, page 18: In the second to last sentence, the authors state that studies with missing outcomes will have their outcomes set to zero with a very high within-study variance so that the substitution for the missing outcome 'will have a negligible effect on the meta-analytic estimate'. If this will have a negligible effect on the meta-analytic estimate by design, then why even include it in the model?

Studies that are missing data on one outcome but have data for another outcome would be included in the multivariate analysis to lend statistical support to studies that measure both outcomes and to increase the precision of the overall estimates.

For reference, please see:

Debray TP, Schuit E, Efthimiou O, Reitsma JB, Ioannidis JP, Salanti G, et al. An overview of methods for network meta-analysis using individual participant data: when do benefits arise? *Stat Methods Med Res*. 2018; 7(5):1351–64.

Jackson D, White IR, Price M, Copas J, Riley RD. Borrowing of strength and study weights in multivariate and network meta-analysis. *Statistical Methods in Medical Research* 2015. doi:10.1177/0962280215611702.

Riley RD, Price MJ, Jackson D, Wardle M, Gueyffier F, Wang J, et al. Multivariate metaanalysis using individual participant data. *Res Synth Methods*. 2015 Jun; 6(2):157–74.

Riley RD, Jackson D, Salanti G, Burke DL, Price M, Kirkham J, et al. Multivariate and network meta-analysis of multiple outcomes and multiple treatments: rationale, concepts, and examples. *BMJ*. 2017 13;358:j3932.

References that are italicized and bolded have been added to the protocol to provide additional background on the concept and conduct of multivariate meta-analysis in the context of IPD-MA.

Assessment of study- and participant-level heterogeneity, page 20: The second paragraph on page 20 ('Based on our review...') does not have a single reference. There are some statements, such as, but not limited to, 'two-stage analyses of interaction effects are subject to ecological bias...' that warrant a reference. Also in this paragraph, for the last sentence it is unclear how the authors plan to assess whether a certain interaction is consistent across studies but not statistically significant within studies; this should be clarified.

We thank the reviewer for their important point here. We have added the following text for clarification and reference:

While two-stage analyses of interaction effects that fail to separate between- and within-study heterogeneity are subject to ecological bias⁹³ and our inference about the importance of interaction terms will primarily be derived from one-stage analyses, we will use a two-stage analysis to compare the magnitude of the interaction effects across studies.

We have added the reference:

93. Fisher DJ, Copas AJ, Tierney JF, et al. A critical review of methods for the assessment of patient-level interactions in individual participant data meta-analysis of randomized trials, and guidance for practitioners. *J Clin Epidemiol* 2011;64(9):949-67. doi:

10.1016/j.jclinepi.2010.11.016 [published Online First: 2011/03/16]

We did not identify another sentence in that paragraph that warranted a reference.

In regards to the reviewer's latter point, we may well find that an interaction is not statistically significant within studies (i.e. when estimated within each study in the first stage of a two-stage analysis), but does reach statistical significance across studies (i.e. when IPD is pooled in a onestage analysis, appropriately accounting for study-level clustering, covariates, and myriad sources of measurement error). Increasing the sample size within important subgroups (i.e. DENV co-infected pregnant women) is part of the stated purpose of the IPD-MA with the idea that clinically important interactions may not be detectable because of the small sample size within certain subgroups (e.g. pregnant women who are co-infected with ZIKV and DENV-4 in the first trimester of pregnancy).

Objective 3, page 21: I have some concerns about the third objective looking at decomposing the total effect of ZIKV infection using causal mediation analysis. The primary objective is to decompose the total effect of ZIKV infection on infant outcomes via mediation through concurrent or prior DENV infection. The assumption behind this analysis is that DENV infection is along one causal pathway from ZIKV infection to infant outcomes (ZIKV→DENV→Infant outcome). However, the authors do not mention any evidence that DENV infection itself could cause an adverse fetal/infant/child outcome. I believe that prior evidence warrants assessing DENV infection as an effect measure modifier, but given the current explanation I am not convinced that it warrants mediation assessment. Additionally

given that there is some evidence of Zika-Dengue interaction due to antibody-dependent enhancement, there may be conceptual issues related to a possible mediator-exposure interaction that are not incorporated into standard approaches of mediation

(<https://www.ncbi.nlm.nih.gov/pmc/articles/PMC5476432/>).

The reviewer makes an excellent point: the the mediator in question needs to cause the outcome to warrant a causal mediation analysis. We included DENV as an example of a potential mediator in Objective 3 and should have been clearer that DENV was a placeholder rather than the effect measure modifier of interest for the effect decomposition that we would conduct, if warranted, for Objective 3. We agree that there would be no reason to decompose the total effect of ZIKV on the outcomes described in this analysis if the posited mediator is not a mediator (i.e. it does not affect the outcome).

We have clarified our language on page 21, which now reads:

If we find evidence in the literature that the effect measure modifier identified in Objective 2 (e.g. DENV) may affect the outcome (e.g. CZS), we will apply inverse probability of treatment weighted-marginal structural models to decompose the total effect of ZIKV on the outcome of interest into the direct effects of ZIKV infection, the effect of ZIKV infection mediated by the posited effect measure modifier, and the effect of the interaction between ZIKV and the effect measure modifier.^{96 97}

We had cited an article that addresses effect decomposition in the presence of interaction:

97. VanderWeele TJ, Tchetgen Tchetgen EJ. Attributing effects to interactions. *Epidemiology* (Cambridge, Mass) 2014;25(5):711-22. doi: 10.1097/EDE.0000000000000096

Selection bias, page 23: The authors mention potential selection bias related to inappropriately adjusting for time-varying confounding. However, there was no mention earlier of assessing ZIKV infection as a time-varying exposure. This should be adding to the section on exposure definitions.

We thank the reviewer for their thoughtful comment. We have added the following text to page 14:

Where possible, ZIKV and other infections (e.g. DENV, CHIKV, STORCH pathogens) will be modelled as time-varying, rather than time-fixed covariates.

We say, “where possible,” here because some studies consisted of two time points (ZIKV test during pregnancy and subsequent fetal, infant, child outcome). In the case of a study with only two time points, ZIKV infection cannot be measured as a time varying exposure.

General comments

Could the authors investigate risk factors for ZIKV infection using the pooled data? There was no mention of this objective in the manuscript. Also, is the consortium open to other proposals for research questions not specified in this manuscript? If so, that should be articulated.

Estimating risk factors associated with fetal ZIKV infection, if possible in a group of studies included in the systematic review, is within the scope of our proposed objectives. Investigating risk factors for maternal ZIKV infection is somewhat outside of the scope of our research questions and would not be addressed as part of the proposed IPD-MA.

Consortium members have not yet reached an agreement or developed a path forward for data sharing beyond the analyses necessary to meet the proposed objectives of the IPD-MA. That continues to be of great interest to some consortium members and we will continue to try and support a path forward for data sharing in the interest of advancing public health. Currently, there is no plan to make the dataset available for additional analyses beyond what is proposed in the protocol but we will revisit that conversation with study principal investigators and active surveillance systems as the IPD-MA moves forward.

The Data sharing statement (page 28) speaks to this:

Not all investigators are willing to share study for analyses beyond what has been proposed here. Governance issues related to sharing the de-identified, participant-level data used in the proposed analyses will be described in the manuscripts that present the results of the proposed analyses.

RESPONSE TO REVIEWER 2

The ethics approval of each participant should be guaranteed by the PI of each study through a certificate they ask each of the parents for the permission.

We will only include studies that provide documentation of ethics review and approval for their individual study. In keeping with standard practice in this IPD-MA, we are not collecting information that could be used by the study team to identify individual research participants. While studies are expected to maintain consent forms and their documentation of the informed consent process is an integral part of their application for ethical approval of their study, the collection of individual consent forms for the IPD-MA would provide us with data that we could use to identify individual research subjects, which we have pledged not to collect for ethics purposes. The World Health Organization Ethics Review Committee and the Emory University Institutional Review Board have deemed this IPD-

MA exempt from consideration as human subjects research because we are using de-identified, previously collected data and those data are being used for the same purposes for which they were originally collected. As such, we do not plan to collect the individual consent forms from research participants, which are required to be collected and stored by individual research teams. We will only collect documentation of ethics review and approval provided by individual studies' ethics review committees or institutional review boards.

It is important to state specifically how the researchers could integrate and/or compare the prediction models or outcomes of each study with those obtained by this IPD study.

While the IPD-MA is uniquely placed to develop and evaluate a risk prediction tool, given their limited sample size, and other issues with sample selection (e.g. limiting enrollment to symptomatic pregnant women), many individual studies may not be well placed to develop and evaluate an externally validated risk prediction tool. While we might qualitatively compare the predictions from our tool to those of other tools that are available at that time, we are unaware of any researchers' proposal to develop a clinical risk prediction model that would be similar to the model that we have proposed here.

We do plan to compare the outcomes of individual studies to the outcomes of the IPD-MA. Summarizing the current evidence, including evidence from both studies that agreed to contribute data and those that did not agree to contribute data to the IPD-MA, is an expected part of an IPD-MA and will certainly be part of the literature review that informs the introduction and discussion in our manuscript. We expect to include both a qualitative and quantitative comparison of published evidence from comparable studies (both those that do and do not contribute data to the IPD-MA) and evidence from the IPD-MA.

Do deserve the serology system used in primary studies more restriction?. Current studies use different home made or commercial techniques, and they could introduce differences in prevalence numbers, mainly in those areas where dengue circulate or where YFV vaccination is mandatory.

The reviewer makes an important point, namely that differences in diagnostic assays and testing procedures (e.g. type of assay, brand, sample tested, sample testing procedures) for ZIKV and closely associated flaviviruses will be an important source of bias in our analyses. We will compare estimates from studies that employed diagnostic assays that are considered to be more accurate to estimates from studies that employed diagnostic assays that are considered less accurate, which would include the assays generally employed at the beginning of the epidemic. We do not plan to exclude studies that deployed low accuracy assays or diagnostic assays of unknown sensitivity and specificity, but will rather account for these differences and clearly elaborate current knowledge on assay sensitivity and specificity and the way that assay accuracy was evaluated (e.g. internal or external investigation of test accuracy).

RESPONSE TO REVIEWER 3

This is a protocol for an individual participant data meta-analysis to estimate the risk of Zika virus infection during pregnancy, examine potential sources of heterogeneity and develop a risk prediction model to identify pregnancies at highest risk of congenital Zika syndrome. It is a collaborative work aiming to include the PIs from all relevant longitudinal studies. The planned analysis is described clearly and in detail and the importance of the review is evident. I have only a few comments, mainly pertaining to the methodology.

Authors discuss the important differences in the distribution of microcephaly cases and aim to identify and explore heterogeneity. What will authors' strategy be in the case of severe heterogeneity? Will they still estimate the overall relative and absolute risk of CZS or will they only present results per subgroup?

In the case of severe heterogeneity, we will present both the subgroup and overall estimates and be clear that the overall estimate must be interpreted with extreme caution because of the excessive level of heterogeneity. In the case of extreme heterogeneity, we may decide to only include overall estimates in the Appendix to help the reader understand that they cannot be interpreted with any certainty.

Authors state that "IPD can be analyzed in either a one- or a two-stage meta-analysis while AD can only be meta-analyzed using a two-stage approach" (Page 13 of 45, lines 24-25). I think that this statement might be misleading as one-stage meta-analysis can also be used when only AD are available. See for example model (2) in the paper by Simmonds and Higgins (Simmonds and Higgins, 2016) where the exact likelihood is used instead of making a normality assumption for the effect estimates.

We thank the reviewer for directing us to this reference and agree that this sentence is inaccurate. We have removed the sentence, "IPD can be analyzed in either a one- or a two-stage metaanalysis while AD can only be meta-analyzed using a two-stage approach."

Authors say that they will use GRADE to compare quality of evidence in Bayesian and frequentist models (Page 25 of 45, lines 41-45). To my understanding, the only model that will be applied both in a Bayesian and a frequentist framework is the multivariate meta-analysis (described in Page 21 of 45, lines 41-end). Could you clarify to which models quality of evidence will be assessed? Also, why the focus is on comparing quality in Bayesian and frequentist models instead of assessing quality of evidence for the key analyses? What would a different assessment depending on the framework imply?

We thank the reviewer for their excellent comment here. We discuss our intention to compare the results of quantitative bias analyses conducted under a Bayesian and frequentist framework in Step 4, page 23. At issue here is that the methods for quantitative bias analyses under the frequentist paradigm may be insufficient to address the myriad, correlated sources of measurement error in this analysis.

To clarify our intention of comparing results from fully Bayesian and frequentist we have added the additional text to Step 4. Quantitative bias analysis:

Given the complexity and level of measurement error, we will conduct a quantitative bias analysis under a Bayesian framework where we use a combination of expert opinion, laboratory EQA, and external and internal assessment of the relative accuracy of diagnostic assays and other methods for cofactor and outcome ascertainment to inform the prior distributions of bias parameters. Where possible, we will apply frequentist methods for quantitative bias analysis¹⁰⁸ as a sensitivity analysis and will use the GRADE criteria¹⁰⁹ to compare the quality of the evidence from Bayesian and frequentist models, with a focus on how imprecision, inconsistency, indirectness, magnitude of effect differ in the Bayesian and frequentist approaches to addressing the myriad sources of bias expected to affect these analyses.

In addition to our discussion of the proposed quantitative bias analysis (pages 23-25), we had included a section on reviewing study quality, which has important implications for the assessment of bias in the overall analysis. This section is on page 17 and is called, "Critical review of study quality."

Considering both of the sections where we describe how we will detect and describe measurement error, we believe that we have sufficiently addressed the need to qualitatively and quantitatively describe the important role of measurement error in the proposed analyses.

Assessing the quality of the evidence for key analyses will be a central component of the results.

Frequentist methods for addressing measurement error and other sources of bias may not be able to encompass the complex, inter-related sources of measurement error. We believe that meaningful differences in the bias analyses conducted under a Bayesian and frequentist framework would indicate the need for further effort to develop Bayesian methods for measurement error in the context of IPD-MA.

Minor comments

Page 19 of 45, line 37: "manager" instead of "manger"

We thank the reviewer for their attention to detail. We have remedied the spelling error mentioned here.

VERSION 2 – REVIEW

REVIEWER	Dr. Jordan Cates ORISE Fellow National Center of Birth Defects and Developmental Disabilities U.S. Centers for Disease Control & Prevention U.S.A. The reviewer works on a Zika cohort study that could potentially be included in this meta-analysis. The conclusions presented in this review are those of the author and do not necessarily reflect the official position of the U.S. Centers for Disease Control and Prevention.
REVIEW RETURNED	19-Apr-2019

GENERAL COMMENTS	Dear editor and authors, I was Reviewer #1 in the original manuscript submission. The authors have clearly addressed all of my comments and incorporated edits where necessary. I appreciate the detailed responses and believe the manuscript has been improved. I have no additional comments.
---